# Complex Formation of Ag^+^ and Li^+^ with Host Molecules Modeled on Intercalation of Graphite

**DOI:** 10.3390/molecules29173987

**Published:** 2024-08-23

**Authors:** Yuriko Uetake, Hiroyuki Takemura

**Affiliations:** Department of Chemical and Biological Sciences, Faculty of Science, Japan Women’s University, Mejirodai 2-8-1, Bunkyo-ku, Tokyo 112-8681, Japan; yuriko.uetake@gmail.com

**Keywords:** intercalation, cation intercaland, cation–Pi, Pi-ligand, pyrene

## Abstract

Pi-stacked and box-shaped host molecules with xanthene as the basis and pyrene as the π-plane were synthesized to verify cation–π interactions between graphene and metal cations. Since crystal structure analysis was not available, DFT calculations were performed to determine the optimized structure, and the π-planes were found to have a slipped parallel structure, with average distances of 456.2–581.0 pm for the stacked compound and 463.4–471.4 pm for the box-shaped compound. Li^+^ and Ag^+^ were chosen as acceptors for complexation with metal ions, and their interactions with the π-plane were clarified by NMR titration. Clearly, the interaction with metal ions increased when pyrene π-planes were stacked rather than the pyrene itself. In the stacked compound, the association constants of Ag^+^ and Li^+^ were similar; however, in the box-shaped host molecule, only Ag^+^ had moderate coordination ability, but the interaction with Li^+^ was very weak, comparable to the interaction with pyrene. As a result, intercalation is more likely to occur in stacked host compound **1**, which has some degree of freedom in the pyrene rings, than in the box-shaped compound.

## 1. Introduction

Since the interaction between π-electrons and alkali metal cations in a vacuum has been measured [1,2], the cation–π interaction has been investigated in detail by gas-phase experiments and theoretical calculations [3,4,5,6]. Many models have been designed and synthesized and their interactions with cations have been investigated in solutions and solids [7,8,9,10]. On the other hand, it is well known that graphite intercalates with metals, metal salts, acids, etc. Complexes of graphene, its constituent unit, with metal ions are currently the subject of basic research on drug delivery [11]. Graphene is also a basic research topic for improving the capacities of Li batteries [12,13,14]. In addition, the permeation of alkali and alkaline earth metal ions through graphene oxide membranes is also being investigated [15,16]. Graphene, the structural unit of graphite, is rich in π-electrons and is a very interesting target for studying cation–π interactions. We previously designed an intramolecular cation–π recognition system and synthesized compounds in which the cation-binding moiety is a crown ether unit, and pyrene and/or coronene are used as the π-planes [17,18]. Using this system, cation–π interactions in a solution can be visualized. The measurement of the association constants of the complexes and comparison of the conformational and energetic changes by DFT calculations revealed that the two units come closer together as a result of the complexation of the cation in the crown part. In the next stage, the authors focused on graphene as the π-plane. Two molecules were designed and synthesized as models for the intercalation of cations in graphite, and the magnitude of cation–π interactions was estimated. Such studies have been reported only in the case of theoretical calculations, and are few in number [19,20,21]. Pyrene is a large π-conjugated aromatic compound with a high fluorescence quantum yield that readily exhibits strong excimer emission. The excimer luminescence of pyrene and its derivatives have been widely used for the analytical detection of various compounds and as a common emission source for sensors, organic light-emitting diodes (OLEDs), and organic photovoltaic cells (OPVs). Although many basic studies have reported the fluorescent properties of pyrene for the development of functional materials, its application as an intercaland model has not been reported. However, several types of layered compounds have been reported, as shown in Figure 1. The effect of the molecular configuration of 1,2-di(pyrenyl)benzene on singlet fission (SF) dynamics was investigated by steady-state and time-resolved spectroscopy [22]. Singlet fission (SF) is a process in which two excited triplet states are formed by a spin-allowed transition from an excited singlet state and has been studied for applications in solar cell photovoltaics. A stacked compound with two pyrenes bonded to the 1- and 8-positions of the naphthalene ring, 1,8-bis(pyren-2-yl)naphthalene (BPyN), has been reported to exhibit remarkable single-molecule excimer emission in solution and films [23]. Li et al. synthesized a compound, **X2P**, in which 1-pyrene units were introduced at the 4- and 5-positions of xanthene, and the distance between the pyrene units was kept constant, and described the π-π interaction of the intramolecular excimer in the crystal [24]. We designed stacked compound **1** and boxed compound **2** as model molecules for a metal-ion sandwich complex formation, referring to these types of stacked compounds (Figure 2).

## 2. Results and Discussion

### 2.1. Preparation of Compounds

Compound **1** is a known compound and was synthesized according to a previously reported method [25]. Pyrene, bis-pinacolato diboron, [Ir(OMe)COD]_2_, and dtbpy were added as catalysts to cyclohexane under Ar and heated and stirred at 80 °C for 14 h. After separation by silica gel chromatography, 2-pyreneboronic acid pinacol ester (2-pyrene-BPin) and pyrene-2,7-diboronic acid pinacol ester (2,7-pyrene-diBPin) were simultaneously obtained in a 30.7% and 22.5% yield, respectively (Figure 1) [26]. Two pyrene boronic acid derivatives obtained from this reaction were used to synthesize compounds **1** and **2**. In the synthesis of **1**, the reaction did not proceed with 2-pyrene-BPin; however, when converted to potassium 2-pyrenyltriolborate and reacted in the presence of potassium carbonate and Pd(PPh_3_)_4_, compound **1** was obtained in 24.3% yield [27]. Compound **2** was obtained in 23.0% yield by the reaction of 2,7-pyrene-diBPin with 4,5-dibromo-2,7-di-*tert*-butyl-9,9-dimethyl-9*H*-xanthene in the presence of K_2_CO_3_ and Pd(PPh_3_)_4_, without using high-dilution conditions.

### 2.2. Optimized Structures by Theoretical Calculations

Because X-ray crystallography was difficult due to the nature of the crystals, DFT calculations (B3LYP/6-31Gd) were performed for **1** and **2** to estimate the optimized structures. *tert*-Butyl and methyl groups were omitted from the calculations.

In the case of compound **1**, the π-planes of the optimized structures were found to be slipped parallel (Figure 3). The intersection angles of the xanthene and pyrene rings were 130.5°and 134.1°, respectively. Therefore, the two pyrene rings were not parallel. The distance between the pyrene–pyrene rings was 456.2–581.0 pm, which was equivalent to the spacing of graphite intercalates. Moreover, pyrene rings are flexible, allowing the inclusion of guests of various sizes. The pyrene ring in compound **2** also had a slipped parallel structure, and the distance between the pyrene rings was 463.4–471.4 pm. The angle between the xanthene skeleton and pyrene ring was 130.1°, which was similar to that of compound **1**, although the pyrene ring was more rigidly fixed than that in compound **1**.

### 2.3. Complexation of Li^+^ and Ag^+^ Ions

Li^+^ and Ag^+^ were chosen as the guest metal cations. Among the alkali metals, Li^+^ was chosen because it is the hardest, has the smallest ionic radius, and is known to interact strongly with π-electrons [3,4,5,6]. Among the transition metals, Ag^+^ was chosen because it is the most popular metal ion and has long been shown experimentally to interact with π electrons [28,29,30,31].

NMR titration (THF-*d_8_*) was performed to investigate the complexation with metal ions (Ag^+^ and Li^+^). The data were analyzed using a nonlinear least-squares fitting method with a variant of the Benesi–Hildebrand equation [32]. K_a_ values were calculated using the change in the chemical shifts in the pyrene ring proton signal. When the host and guest form a 1:1 complex, the chemical shift changes induced by the complexation (Δ*δ_obs_*) can be expressed by Equation (1). For each plot, the binding constants were obtained by curve fitting using Kaleida Graph™ by applying Equation (1).
(1)$${\Delta \delta }_{obs}=\frac{\Delta \delta }{2K{\left[H\right]}_{0}}\left[1+K{\left[G\right]}_{0}+K{\left[H\right]}_{0}-{\left\{{\left(1+K{\left[G\right]}_{0}+K{\left[H\right]}_{0}\right)}^{2}-4K^{2}{\left[G\right]}_{0}{\left[H\right]}_{0}\right\}}^{\frac{1}{2}}\right]$$

In Equation (1), fitting is possible only when the [Host]/[Guest] ratio of the complex is 1:1. For reconfirmation, a job plot was also constructed for the complexation of **1** with Ag^+^ ions, and indeed a 1:1 complexation was observed. The results are shown in Table 1. Fitting of the data points obtained as a result of titration yielded good R values (see Appendix A).

When Ag^+^ was added to the pyrene solution, all the signals of the aromatic rings shifted to a lower field. With compound **1** and Ag^+^, some signals were high-field-shifted and some were low-field-shifted depending on the proton. In the case of compound **2** and Ag^+^, the chemical shift was very small, but shifted to higher fields. However, when Li^+^ was added to pyrene or host molecules **1** and **2**, the proton signal of the host shifted to higher fields in both cases. These phenomena are probably due to the conformational changes in hosts **1** and **2** upon the inclusion of Li^+^ and Ag^+^ ions, the different interaction sites, and the different nature of their interactions. The main interaction of Li^+^ with the π-plane is a cation–π interaction; however, in the case of Ag^+^, in addition to the cation–π interaction, the *d*-*π** interaction caused by the overlap between the *d* orbital of Ag^+^ and the π* orbital of the π-plane is considered to be significant. The titration results clearly showed that the interaction with metal ions increased for pyrene-ring-layered compounds **1** and **2** rather than for pyrene itself. In stacked compound **1**, the K_a_ values for Ag^+^ and Li^+^ were similar, but the ability of **1** to form complexes with Li^+^ was much higher than that of compound **2** or pyrene. In contrast, box-shaped host molecule **2** had a relatively strong coordination ability only for Ag^+^, and the interaction with Li^+^ was as weak as that of pyrene. In the case of compound **1**, this was probably due to the π-electron clustering effect of the two pyrene rings and the fact that the host could flexibly change its interlayer distance to sandwich the guest (Ag^+^ and Li^+^). Conversely, the π-plane of the box-shaped host molecule **2** is rigidly fixed, and the solvated Li^+^ ion, which is bulky, has a large solvation force; therefore, extra energy is required for the Li^+^ ion to desolvate and then interact with **2**, which causes the pyrene rings to be less likely to interact with the Li^+^ ion. In contrast, in the case of Ag^+^, the system may be stabilized by sandwiching between two π-planes that are maintained at a distance suitable for *dπ-pπ** interactions. Furthermore, the strength and selectivity of the interaction between Li^+^, Ag^+^, and π electrons can be explained based on the HSAB theory. Information on the crystal structures of the complexes of Li^+^ and Ag^+^ with **1** or **2** is of great interest for understanding the results of NMR experiments, but all attempts to isolate the complexes in the presence of large excesses of metal ions with compounds **1** and **2** have failed. Yáñez et al. used theoretical calculations to determine the position of Li^+^ ion in complexes of PAHs and Li^+^ and showed that there are several structures for the energy minima of Li^+^-pyrene complexes [33] In the case of our ligand, the system is more complex, and calculations could not be performed; however, it can be inferred that there are several stable structures for Li^+^-**1** and Li^+^-**2**.

## 3. Materials and Methods

### 3.1. Experimental Section

#### 3.1.1. General Procedure

Melting points were obtained using a Yanaco MP-500D apparatus in Ar-sealed tubes and were uncorrected. NMR spectra were collected on a JEOL AL-300 spectrometer (300.4 MHz for ^1^H, 75.6 MHz for ^13^C) with TMS or solvents as internal references. FAB-MS data were collected on a JEOL JMS-SX/SX102A instrument. Silica gel PTLC was performed using PF_254_ (Merck). All reagents were commercially available and used as supplied without further purification.

#### 3.1.2. Synthesis of Compound **2** from 2,7-bis(4,4,5,5-tetramethyl-1,3,2-dioxaborolan-2-yl)pyrene

In a 100 mL three-necked flask, toluene (40 mL), 2 M aq. K_2_CO_3_ (4 mL), ethanol (10 mL), 2,7-pyrene-diBPin (0.20 g, 0.44 mmol), 4,5-dibromo-2,7-di-*tert*-butyl-9,9-dimethyl-9*H*-xanthene (0.21 g, 0.46 mmol), and Pd(PPh_3_)_4_ (0.050 g, 0.043 mmol) were added and bubbled using Ar gas for 20 min. The mixture was then heated and stirred at 90 °C for two days. The precipitates produced were filtered by suction and dissolved in CH_2_Cl_2_, and the solution was filtered through a celite pad. Toluene (10 mL) was added to the filtrate and concentrated to obtain pure **2** as a white powder (0.052 g, 23.0%). M.p. > 460 °C. ^1^H NMR (300 Hz, CDCl_3_) *δ* 1.44 (s, 36H), 1.89 (s, 12H), 7.07 (s, 8H), 7.45 (s, 4H), 7.56 (s, 4H), 7.93 (s, 8H). ^13^C NMR (300 Hz, CDCl_3_) *δ* 31.67, 33.21, 34.68, 34.97, 76.59, 77.02, 77.22, 77.44, 122.35, 123.49, 126.04, 126.44, 126.98, 129.48, 129.61, 130.52, 135.46, 145.36, 145.57. HRMS, calcd for C_78_H_73_O_2_ (M+H)^+^ = 1041.5614. Found, 1041.5599.

#### 3.1.3. Silver Ion Complexation with **1**: Preparation of Solutions for ^1^H NMR Titration

Preparation of host solution (1): Dissolve **1** (0.72 mg, 2 × 10^−3^ mmol) in 1.0 mL of THF-*d_8_* to make a solution of 2.0 × 10^−3^ mol/L.Preparation of guest (AgClO_4_) solution (2): AgClO_4_ (12.4 mg, 0.06 mmol) was dissolved in 0.30 mL of THF-*d_8_* to make a 0.20 mol/L solution.Preparation of guest (AgClO_4_) solution (3): AgClO_4_ (0.25 g, 1.2 mmol) was dissolved in 0.60 mL THF-*d_8_* to make a solution of 2.0 mol/L.

Solution (1) (0.30 mL) was placed in NMR tubes, and solution (2) was added to a constant volume (0–80 μL) using a microsyringe. In other NMR tubes, solution (3) was added in 10 μL increments using a microsyringe to prepare a mixed solution. THF-*d_8_* was then added until the total volume was 0.60 mL and mixed well. The solution was prepared such that the [H]/[G] ratio was 0–120. ^1^H NMR was performed, and the chemical shifts (pyrene) were recorded. Equation (1), a modification of the Benesi–Hilderand method, was used for the analysis.

The obtained titration curve was analyzed by the nonlinear least square method using Kaleida Graph^TM^ for equation to obtain the binding constant (39.3 ± 9.7 L mol^−1^). The following complexation experiments between Ag^+^, Li^+^, and pyrene, compounds **1** and **2** were performed in the same way.

#### 3.1.4. Job Plot

A THF-*d_8_* solution of compound **1** (0.02 mol/L) and a solution of AgClO_4_ (0.02 mol/L) were prepared. These solutions were added to the NMR tubes in 0.02–0.18 mL increments to give [host] + [guest] = 0.20 mL.

## 4. Conclusions

Compounds **1** and **2** were synthesized to investigate the complexation of metal cations with layered compounds as a model for the intercalation of graphite and cations. Through computer-assisted optimization, compound **1** was found to have a structure in which the π-planes are in a slipped parallel structure, and the distance between the π-planes is relatively flexible. In the case of compound **2**, the pyrene rings can move in parallel, but the distance is not variable. Pi-plane dimerization is generally more favorable in the slipped parallel form than in the fully overlapped form. Therefore, the misalignment of the pyrene rings in **1** and **2** may be caused by the repulsion of the π-electron dispersion forces. Interactions with the metal ions (Li^+^ and Ag^+^) were measured by ^1^H NMR titration. When AgClO_4_ and LiClO_4_ were mixed, the NMR titration results showed 1:1 complexation, which led to the conclusion that the metal ions were encapsulated between the two pyrene rings. Another feature is that the association constant varies depending on the structure of the compound. These results indicate that not only does metal intercalate, which is a process already in practical use, but cations can also intercalate between the graphite layers.

## Data Availability

Data are contained within the article and Appendix A.

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
