# Peer review of "Complex Formation of Ag+ and Li+ with Host Molecules Modeled on Intercalation of Graphite"

_molecules, 2024, doi:10.3390/molecules29173987_

Round 1

Reviewer 1 Report

Comments and Suggestions for Authors

The manuscript describes the synthesis of a pyrene-substituted xanthene derivative as a model for graphite and investigates its binding with Li and Ag cations. The primary concern is that the DFT calculations are performed only for the ligands, which is somewhat trivial. Instead, calculating the barrier of pyrene rotation would provide deeper insights into this system. Additionally, DFT models for the corresponding Li and Ag complexes would be beneficial in rationalizing the NMR studies. Therefore, I recommend acceptance pending major revisions.

Minor comments:

1. Please briefly explain why Li+ and Ag+ were chosen.

2. The shimming for 1H NMR in Figure S10 is bad; please remeasure the spectra since chemical shift has a direct impact on the results.

3. Line 144, “the complexation ability of Li+ was much greater than that of the other hosts” the comparison between Li+(guest) and hosts is not valid. Please fix it.

Author Response

Comments and Responces

Reviewer (1) Commentas and Responces 

The manuscript describes the synthesis of a pyrene-substituted xanthene derivative as a model for graphite and investigates its binding with Li and Ag cations. The primary concern is that the DFT calculations are performed only for the ligands, which is somewhat trivial. Instead, calculating the barrier of pyrene rotation would provide deeper insights into this system. Additionally, DFT models for the corresponding Li and Ag complexes would be beneficial in rationalizing the NMR studies. Therefore, I recommend acceptance pending major revisions.

In response to the comment "Calculating the rotational barrier of the pyrene instead would provide greater insight into this system," it is possible that the two pyrene units can rotate, but we did not measure this because the rotation of the pyrene is not relevant for the purposes of this paper. It provides information about the structure, but cannot be related to the interaction with the cation. The only insight that matters for this study is the binding constant of the metal ion with the π-planes facing each other.

The reviewer is right about the point that "DFT models of the corresponding Li and Ag complexes would help streamline NMR studies." The authors are also very interested in these complexes and have tried experiments to isolate them, but all failed. We have added this to the main text. Reviewer 2 made a similar comment, to which we responded: "Theoretical calculations to determine the location of metal ions in the complexes have also been performed by experts in theoretical calculations of PAHs. This literature has been newly added to the list. For example, several stable structures have been calculated for pyrene and Li+ complexes. The fact that pyrene has multiple stable structures clearly means that there are more complicated cases for our ligand, and such theoretical calculations are beyond our capabilities. Moreover, as the title suggests, the aim of this paper begins with the question of whether it is possible to form stable complexes by sandwiching cations in large π systems, and we are confident that this aim can be achieved even if the structure cannot be determined." The authors are not experts in theoretical calculations and the survey itself could be the subject of another theoretical paper. We do not feel the need to do so, at least not in this short paper.

Minor comments:

  1. Please briefly explain why Li+ and Ag+ were chosen.

Typical cations to be compared and verified as guests in exploring cation-pi interactions are alkali metals, which simply have a charge, and transition metals, which have a d-electron that may interact with the π electron. Among the alkali metals, Li+ was chosen because it is the hardest, has the smallest ionic radius, and is known to interact strongly with π-electrons. Among the transition metals, Ag+ was chosen because it is the most popular metal ion that has long been known experimentally to interact with  π-electrons. These explanations and relevant references have been added to the revised text.

  1. The shimming for 1H NMR in Figure S10 is bad; please remeasure the spectra since chemical shift has a direct impact on the results.

Indeed, in this experiment, the shim adjustment did not go well owing to the condition of the NMR instrument, but this did not affect the chemical shift in any way. In fact, according to the titration results, the accuracy of the fitting is R = 0.998; therefore, there is no problem with the measurement results. Even if the results gave better appearance after re-measurement, we believe that the results will remain the same.

  1. Line 144, “the complexation ability of Li+ was much greater than that of the other hosts” the comparison between Li+(guest) and hosts is not valid. Please fix it.

This is a misleading expression. This was rewritten as follows: ‘In stacked compound 1, the Ka values for Ag+ and Li+ are similar, but the ability of 1 to form complexes with Li+ was much higher than that of compound 2 or pyrene.’

Reviewer 2 Report

Comments and Suggestions for Authors

In this paper the authors synthesized two pyrene-based dimer molecules bridged by xanthene. The interactions of these molecueles with Li+ and Ag+ are investigated through H NMR titration. The originality and significance of the results presented in this work are suitable to be published in Molecules, however, major revisions are required before the acceptance of the manuscript.

1. Referencing more relevant previous papers by other researchers is highly recommended. More literatures should be introduced to introduce background, concepts, significance, and applications of presented results.

2. The title may be very misleading to the authors. They need to carefully use "graphene" and "graphite", as they never used these materials.

3. The titration method using NMR is one of the main results of the paper, but no figures related to NMR are shown in the main manuscript. More figures and discussions should be included in the manuscript.

4. The authors should at least present structural analysis data using X-ray source (e.g. XRD), for more detailed structural analysis of the molecule-ion complex.

Comments on the Quality of English Language

The quality of English language is fine to be published.

Author Response

Comments and Responces

Reviewer (2) Commentas and Responces 

  1. Referencing more relevant previous papers by other researchers is highly recommended. More literatures should be introduced to introduce background, concepts, significance, and applications of presented results.

New references on cation-π and graphene-metal ions have been added to the literature. Note, however, that in our search, we found only a few metal ion intercalation papers that are closely related to this paper. We have added commentary on the newly added references.

  1. The title may be very misleading to the authors. They need to carefully use "graphene" and "graphite", as they never used these materials.

As the subject of this paper was to mimic graphite intercalation, there should be no problem with the use of graphite in the title. The authors checked that the terms graphene and graphite were used appropriately in the text.

  1. The titration method using NMR is one of the main results of the paper, but no figures related to NMR are shown in the main manuscript. More figures and discussions should be included in the manuscript.

Figure 1 has been added because it is easier to understand the purpose of the text if the related substances (references, 9, 10, 11) of the compounds synthesized in this study are shown.

It seems that the reviewer thinks that just including the titration results in the Supplementary Information is not enough. However, titration experiments are classical and simple, and the subject of discussion is the results, not the method. The author has not seen such titration results figures in the main text of other papers with similar content. Especially in the case of our short paper, they should be excluded from the main text. We added a note that the results of the titration experiment in the main text should be referred to in the Supplementary Information.

  1. The authors should at least present structural analysis data using X-ray source (e.g. XRD), for more detailed structural analysis of the molecule-ion complex.

It is very interesting to ascertain the structure of molecular ion complexes. However, all attempts to isolate this complex failed. It is clear from the measurement of the association constants that the complex is present in small quantities only when there is a large excess of ligand or metal ions. In my long experience of studying host-guest chemistry, I have to say that we would be very lucky if these complexes could be isolated and measured. In this regard, it is noted in the text that the isolation of the complex was unsuccessful. Theoretical calculations to determine the position of metal ions in complexes have also been performed by experts in theoretical calculations for PAHs. This literature is newly added to the list. For example, several stable structures have been calculated for pyrene and Li+ complexes. The fact that there are several stable structures of pyrene clearly means that there will be more complicated cases for our ligands, and such theoretical calculations are beyond our capabilities. Furthermore, as the title suggests, the purpose of this paper started with the question of whether or not stable complexes can be formed by sandwiching a cation in a large pi system, and we are convinced that we can fulfil this purpose even if the structure cannot be determined.

Round 2

Reviewer 1 Report

Comments and Suggestions for Authors

The revised manuscript is suitable for publication.

Reviewer 2 Report

Comments and Suggestions for Authors

The reviewers properly addressed the comments suggested. The manuscript is ready to be published in present form.